# Programmable frequency-bin quantum states in a nano-engineered silicon device

Marco Clementi [1,7,12] ✉, Federico Andrea Sabattoli[1,8,12], Massimo Borghi[1], Linda Gianini[2,3], Noemi Tagliavacche [1], Houssein El Dirani[3,9], Laurene Youssef [4,10], Nicola Bergamasco[1], Camille Petit-Etienne [4], Erwine Pargon[5], J. E. Sipe[6], Marco Liscidini[1], Corrado Sciancalepore[3,11], Matteo Galli [1] ✉ & Daniele Bajoni [2]

Photonic qubits should be controllable on-chip and noise-tolerant when transmitted over optical networks for practical applications. Furthermore, qubit sources should be programmable and have high brightness to be useful for quantum algorithms and grant resilience to losses. However, widespread encoding schemes only combine at most two of these properties. Here, we overcome this hurdle by demonstrating a programmable silicon nano-photonic chip generating frequency-bin entangled photons, an encoding scheme compatible with long-range transmission over optical links. The emitted quantum states can be manipulated using existing telecommunication components, including active devices that can be integrated in silicon photonics. As a demonstration, we show our chip can be programmed to generate the four computational basis states, and the four maximally-entangled Bell states, of a two-qubits system. Our device combines all the key properties of on-chip state reconfigurability and dense integration, while ensuring high brightness, fidelity, and purity.

Photons serve as excellent carriers of quantum information. They have long coherence times at room temperature, and are the inescapable choice for broadcasting quantum information over long distances, either in free space or through the optical fiber network. Quantum state initialization is a particularly important task for photonic qubits, since adjusting entanglement after emission is nontrivial. Initialization strategies depend on the degree of freedom used to encode quantum information, and the most common choice for quantum communication over optical channels is time-bin encoding[1]. Here, the two-qubit levels consist of the photon being in one of two time windows, generally separated by a few nanoseconds. Time-bin encoding is extremely resilient to phase fluctuations resulting from thermal noise in optical fibers, with qubits maintaining their coherence even over hundreds of kilometers[2,3]. However, the control of the state in which time-bin-entangled photons are generated is challenging, and impractical in emerging nano-photonic platforms. For on-chip manipulation of qubit states, dual-rail encoding, in which the two states of a qubit correspond to the photon propagating in one of two optical waveguides, is a superior strategy[4,5], and is thus a common choice for quantum computing and quantum simulation in integrated

[1]Dipartimento di Fisica, Università di Pavia, Via Agostino Bassi 6, 27100 Pavia, Italy. [2]Dipartimento di Ingegneria Industriale e dell'Informazione, Università di Pavia, Via Adolfo Ferrata 5, 27100 Pavia, Italy. [3]Univ. Grenoble Alpes, CEA-Leti, 38054 Grenoble, France. [4]Univ. Grenoble Alpes, CNRS, LTM, 38000 Grenoble, France. [5]Univ. Grenoble Alpes, CNRS, CEA/LETI-Minatec, Grenoble INP, LTM, 38054 Grenoble, France. [6]Department of Physics, University of Toronto, 60 St. George Street, Toronto, ON M5S 1A7, Canada. [7]Present address: Photonic Systems Laboratory (PHOSL), École Polytechnique Fédérale de Lausanne, 1015 Lausanne, Switzerland. [8]Present address: Advanced Fiber Resources Milan S.r.L., Via Federico Fellini 4, 20097 San Donato Milanese, MI, Italy. [9]Present address: LIGENTEC SA, 224 Bd John Kennedy, 91100 Corbeil-Essonnes, France. [10]Present address: Univ. Limoges, CNRS, IRCER, UMR 7315, 87000 Limoges, France. [11]Present address: SOITEC SA, Parc technologique des Fontaines, Chemin des Franques, 38190 Bernin, France. [12]These authors contributed equally: Marco Clementi, Federico Andrea Sabattoli. ✉e-mail: marco.clementi@epfl.ch; matteo.galli@unipv.it

platforms. Yet this approach is not easily compatible with long-distance transmission links using either optical fibers or free space channels.

Recently, frequency-bin encoding has been proposed, and experimentally demonstrated, as an appealing strategy that can combine the best characteristics of time-bin and dual-rail encodings[6-11]. In this approach, quantum information is encoded by the photon being in a superposition of different frequency bands. Frequency bins can be manipulated by means of phase modulators, and are resistant to phase noise in long-distance propagation. Pioneering studies have investigated the generation and manipulation of frequency-bin-entangled photons in integrated resonators. They have considered quantum state tomography of entangled photon pairs[12], qudit encoding[13], and multi-photon entangled states[14]. The experimental results have all been achievable thanks to the recent development of high-Q integrated resonators in the silicon nitride and silicon oxynitride platforms.

Despite all this progress, there are obstacles that must be overcome in order to exploit the full advantage of photonic integration. In frequency-bin encoding today, the generation of photon pairs occurs via spontaneous four-wave mixing in a single ring resonator, with the desired state obtained outside the chip, by means of electro-optical modulators and/or pulse shapers. And since commercial modulators have limited bandwidth, the frequency span separating the photons cannot exceed a few tens of gigahertz, which sets a limit to the maximum free spectral range of the resonator. Finally, because spontaneous four-wave mixing efficiency scales quadratically with the resonator free spectral range[15], there is also a significant trade-off between the generation rate and the number of accessible frequency bins.

In this work, we show that these limitations can be overcome by utilizing the flexibility of light manipulation in a nano-photonic platform and the dense optical integration possible in silicon photonics. Our approach is based on constructing the desired state by direct, on-chip control of the interference of biphoton amplitudes generated in multiple ring resonators that are coherently pumped. States can thus be constructed "piece-by-piece" in a programmable way, by selecting the relative phase of each source. In addition, since the frequency-bin spacing is no longer related to the ring radius, one can work with very high-finesse resonators, reaching megahertz generation rates. These two breakthroughs, namely high emission rates in combination with high values of the free spectral range, together with output state control using on-chip components, are only possible using multiple rings: they would not be feasible were the frequency bins encoded on the azimuthal modes of a single resonator.

We demonstrate that with the very same device one can generate all superpositions of the $|00\rangle$ and $|11\rangle$ states or, in another

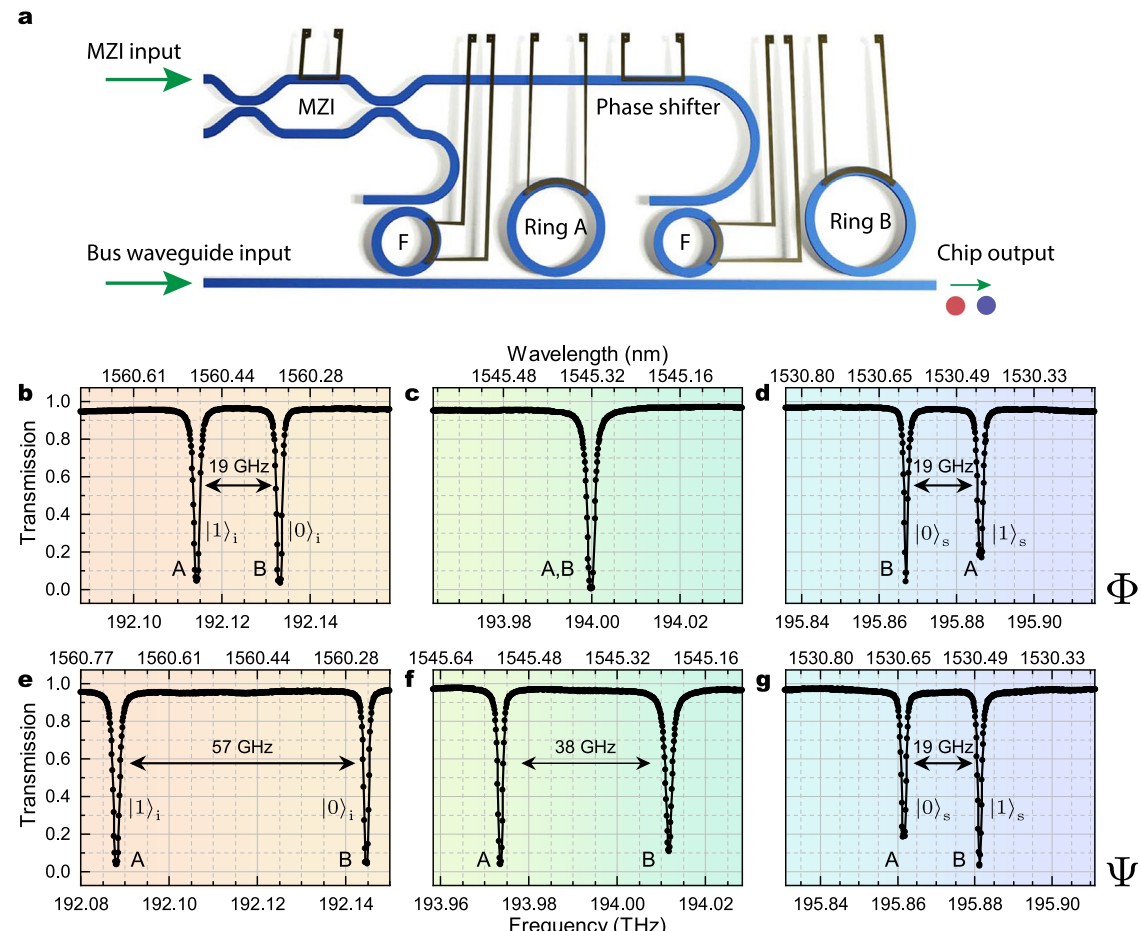

**Fig. 1 | Device layout and transmission spectra. a** Schematic of the device, in which a Mach Zehnder Interferometer (MZI) is used to route optical pumping power to the two generating rings (Ring A and Ring B) via two add-drop filters (F). The pump relative phase for the two rings is controlled by a thermo-electric phase shifter. **b–d** Linear characterization of the sample through the bus waveguide, with the device operated in configuration Φ. A detail of the transmission spectrum around the idler (panel **b**, $m = -5$), pump (panel **c**, $m = 0$), and signal (panel **d**, $m = +5$) bands shows resonances belonging to both ring resonators, identified by labels A and B, respectively. In this configuration, Ring B is associated with the $|0\rangle_{s,i}$ frequency bins for both signal and idler, while Ring A is associated with the $|1\rangle_{s,i}$ resonances for both signal and idler. **e–g** Same as panels **b–d**, respectively, but with the device set in configuration Ψ. Here, Ring A corresponds to the $|0\rangle_s$ resonance for the signal and $|1\rangle_i$ resonance for the idler, Ring B corresponds to the $|1\rangle_s$ resonance for the signal and $|0\rangle_i$ resonance for the idler.

configuration with different frequency-bin spacing, all superpositions of the $|01\rangle$ and $|10\rangle$ states. One needs only to drive the on-chip phase shifter and set the pump configuration appropriately. This means that all four fully-separable states of the computational basis and all four maximally entangled Bell states ($|\Phi^{\pm}\rangle = (|00\rangle \pm |11\rangle)/\sqrt{2}$ and $|\Psi^{\pm}\rangle = (|01\rangle \pm |10\rangle)/\sqrt{2}$) are accessible. Our high generation rate allows us to perform quantum state tomography of all these states, reaching fidelities up to 97.5% with purities close to 100%.

## Results

### Device characterization and principle of operation

The device is schematically represented in Fig. 1a. The structure is operated by exploiting the fundamental transverse electric (TE) mode of a silicon waveguide, with a $600 \times 220$ nm$^2$ cross section, buried in silica. Two silicon ring resonators (Ring A and Ring B) in all-pass configuration act as sources of photon pairs. Their radii are some 30 μm to ensure high generation rates, and they are not commensurate so that the two free spectral ranges are different: FSR$_A$ = 377.2 GHz and FSR$_B$ = 373.4 GHz, respectively. The two rings are critically coupled to a bus waveguide and their resonance lines can be tuned independently by means of resistive heaters. The device also contains a tunable Mach-Zehnder interferometer (MZI), whose outputs are connected to the input of two tunable add-drop filters that allow one to control the field intensity and relative phase with which Ring A and Ring B are pumped in the spontaneous four-wave mixing experiment[16].

Linear transmission measurements through the bus waveguide are shown in Fig. 1b–g. In a first configuration (Fig. 1b–d), which we will later refer to as "Φ", two resonances of Ring A and Ring B are spectrally aligned to be later used for pumping, thus only one transmission dip is observed at 194 THz (1545 nm) in Fig. 1c. Since Ring A and Ring B have different free spectral ranges, the other resonances are not aligned, and one observes double dips, with spacing $\Delta(m) = |m|(\text{FSR}_A - \text{FSR}_B)$, with $m$ being the azimuthal order with respect to the pump resonance. In Fig. 1b and d we plot the transmission double dip corresponding to $m = -5$ and $m = +5$, named "idler" and "signal", respectively. For both the signal and idler bands the resonances of Ring A and Ring B are separated by $\Delta = 19$ GHz. Later, the two frequencies will be used to encode the two states of the qubits, with signal and idler pairs of frequencies representing the two qubits. For this reason, in Fig. 1b and d, we name $|0\rangle_{s,i}$ the two frequency bins closer to the pump, and $|1\rangle_{s,i}$ the two bins further away from the pump, in line with previous works on frequency-bin entanglement[6]. Our device can also operate in a different configuration, which we will refer to as "Ψ". Here Ring A and Ring B are thermally tuned so that the resonances corresponding to the states $|0\rangle_i$ and $|1\rangle_s$ belong to Ring B and those corresponding to $|0\rangle_s$ and $|1\rangle_i$ belong to Ring A (see Fig. 1e–g). As can be seen from all panels in Fig. 1b–g, the resonances of the two generating rings have quality factors $Q \approx 150,000$ (Full width at half maximum $\Gamma \approx 1.3$ GHz), which guarantee well-separated frequency bins and high generation rates.

The basic principle of operation of the device is the following: (i) Ring A and Ring B are set in the proper configuration (e.g., Φ) by controlling the thermal tuners; (ii) The pump power is coherently distributed between the two rings with the required relative phase and amplitude set either through the MZI or directly through the bus waveguide; (iii) Photon pairs are collected in the bus waveguide, with the desired state resulting from a coherent superposition of the two-photon states that would be generated by each ring separately.

### Spontaneous four-wave mixing

The photon generation efficiency through spontaneous four-wave mixing (SFWM) was assessed for the two rings by setting the device in configuration Ψ, which is convenient to pump each ring individually through the bus waveguide. The two resonators were pumped with an external tunable laser, and the chip output was separated in the signal (194.7–197.2 THz), pump (192.2–194.7 THz), and idler (189.7–192.2 THz)

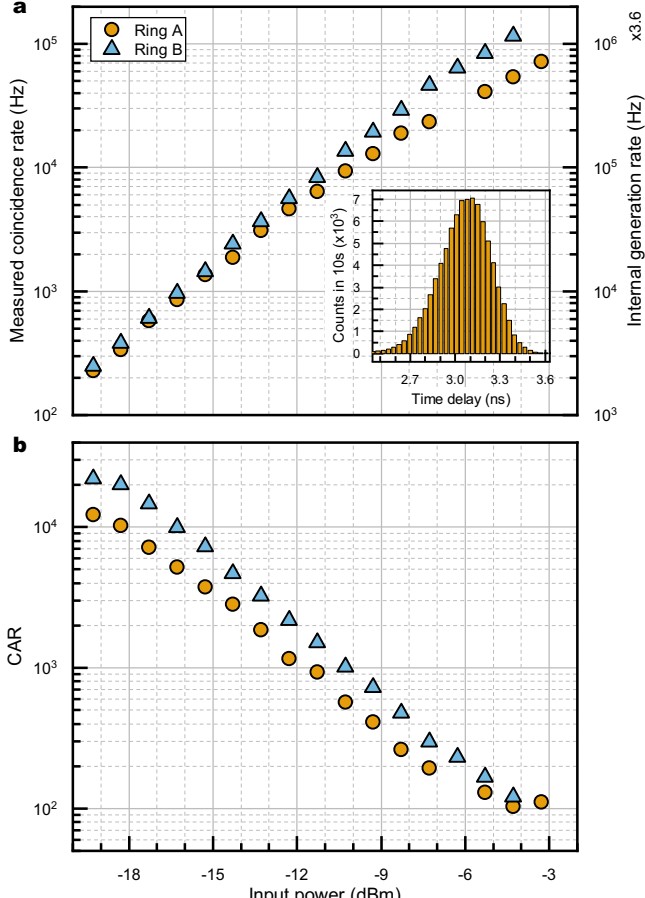

**Fig. 2 | Spontaneous four-wave mixing.** Generation of pairs through spontaneous four-wave mixing using the two rings of the device. The two sets of resonances are shifted such that all the resonances are separated (configuration Ψ). A tunable laser is tuned on resonance with either Ring A or Ring B, and the related signal and idler photons are detected. Similar coincidence rates (**a**) are observed, proving that the two rings have similar generation efficiencies. Inset shows an example histogram of the photon arrival time delays. Panel **b** shows the calculated CAR, which exhibits the typical reduction for the higher values of the input power due to the generation of higher-order photon states.

bands by means of a telecom-grade coarse wavelength division multiplexer (see Supplementary Fig. 1). The generated signal and idler photons were then narrowband filtered using tunable fiber Bragg gratings with an 8 GHz stop-band, and routed to a pair of superconductive single-photon detectors. The overall insertion losses from the bus waveguide to the detectors are 6 and 7 dB for signal and idler channels, respectively. The results of the experiment are summarized in Fig. 2. The two rings exhibit similar generation efficiency $\eta = R/P_{\text{wg}}^2$, with $\eta_A = 57.6 \pm 2.1$ Hz/μW$^2$ for Ring A and $\eta_B = 62.4 \pm 1.7$ Hz/μW$^2$ for Ring B[15]. The internal pair generation rate $R$ can exceed 2 MHz for both ring resonators (Fig. 2a). A high coincidence-to-accidental ratio (CAR) exceeding $10^2$ was obtained for any value of the input power, a necessary condition to ensure a high purity of the generated state (Fig. 2b).

We now turn to the spectral properties of the generated photon pairs and the demonstration of entanglement. We set our device to operate in the Φ configuration, which will later be used to generate the maximally entangled state

$$|\Phi(\theta)\rangle = \frac{|00\rangle + e^{i\theta}|11\rangle}{\sqrt{2}}, \tag{1}$$

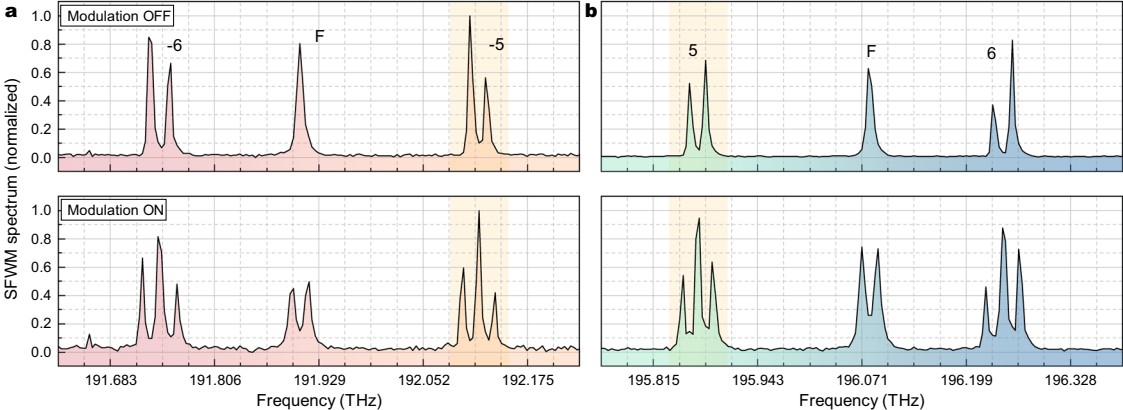

**Fig. 3 | Effect of modulation on spontaneous four-wave mixing spectra.** Normalized spontaneous four-wave mixing spectra for the **a** idler and **b** signal channels after demultiplexing both in the absence (upper panels) and presence (lower panels) of modulation. The bin pair order $m$ with respect to the pump add-drop filter rings is marked, while spontaneous four-wave mixing generated in the add-drop filter rings is marked as F. Note that, despite the different out-coupling efficiency for each resonance and the limited resolution of the spectrometer, it is still possible to observe the expected symmetry in the intensity of the generated bins, and how the bin spacing increases with the azimuthal order $m$. Lower panels show the effect of the double-sideband suppressed-carrier modulation on the signal and idler spectra, where only the first-order sidebands are preserved. The spectra shown here are associated with generation of the state described by Eq. (1), where we chose $\theta = \pi$ (Bell state $|\Phi^-\rangle$). Analogous spectra are attainable for any of the device configurations discussed in this work.

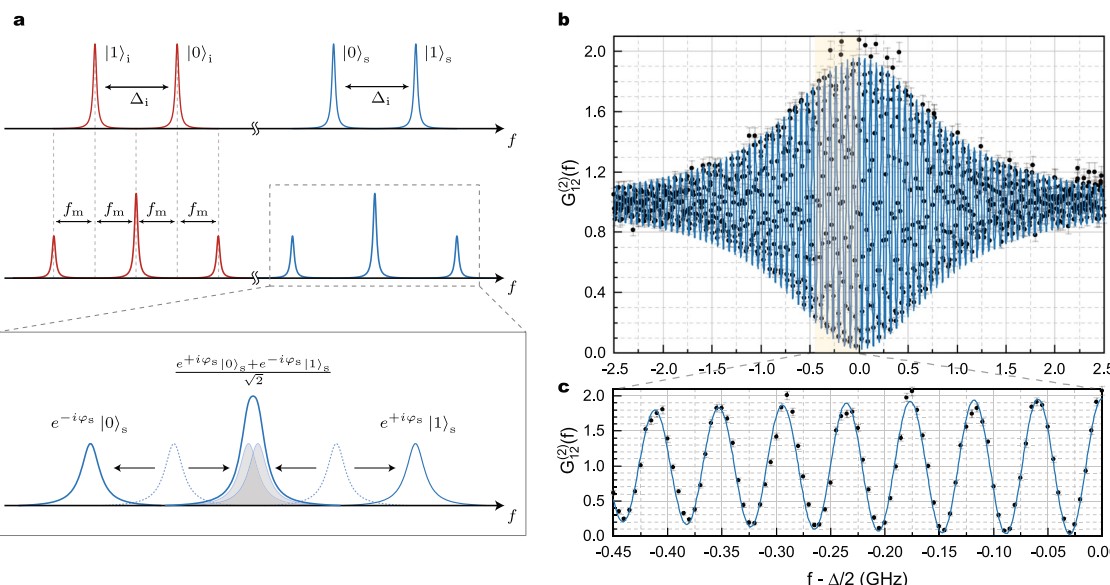

**Fig. 4 | Frequency mixing and two-photon interference. a** Schematic of the effect of modulation on the generated idler (red) and signal (blue) frequency bins. The frequency mixing produces maps each of the signal and idler states in a superposition of three frequency components: the outermost ones are reminiscent of the probability amplitude proportional to $|0\rangle_{s,i}$ or $|1\rangle_{s,i}$, while the "central" bin results in a superposition of the two. Each frequency-shifted bin also acquires a phase $\pm \varphi_{s,i}$ due to the modulation. The superposition of the generated bins is regulated by the modulation frequency, and the overlap is ideally maximized when $f_m = \Delta/2$, when perfect indistinguishability of the generated bins is achieved. **b** Two-photon correlation $G_{1,2}^{(2)}$ of the frequency-mixed bins as a function of the detuning $f_m - \Delta/2$. The experimental points (black dots) were obtained by counting coincidences between frequency-mixed bins at varying modulation frequency, while keeping fixed the modulation phase, and normalizing. Error bars (light gray) were estimated assuming Poissonian statistics. Blue curve represents the best-fit of the curve according with Eq. (2), showing good agreement (**c**) with theoretical predictions.

where $|00\rangle = |0\rangle_s |0\rangle_i$, $|11\rangle = |1\rangle_s |1\rangle_i$, and the phase $\theta$ can be adjusted by acting on the thermo-electric phase shifter after the interferometer (see Supplementary Note 1); $\theta = 0$ and $\theta = \pi$ correspond to the well-known Bell states $|\Phi^+\rangle$ and $|\Phi^-\rangle$, respectively. The corresponding SFWM spectrum of the signal and idler bands is shown in Fig. 3a and b (upper panels); the device was electrically tuned to set $\theta = 0$, with the pump power split equally between Rings A and B by means of the MZI. Here we focus on the azimuthal order $m = \pm 5$, with the generated frequency bins clearly distinguishable in the marginal signal and idler spectra.

## Two-photon interference

In order to demonstrate entanglement, the demultiplexed signal and idler photons were routed (see Supplementary Fig. 1) to two intensity electro-optic modulators (EOMs), coherently driven at $f_m = 9.5$ GHz, which corresponds to half the frequency-bin separation of the selected azimuthal order $m = \pm 5$. The modulators operate at the minimum transmission point (i.e., at bias voltage $V_\pi$) to achieve double-sideband suppressed-carrier amplitude modulation. The amplitude of the modulating RF signal was chosen to maximize the transferred power from the carrier to the first-order

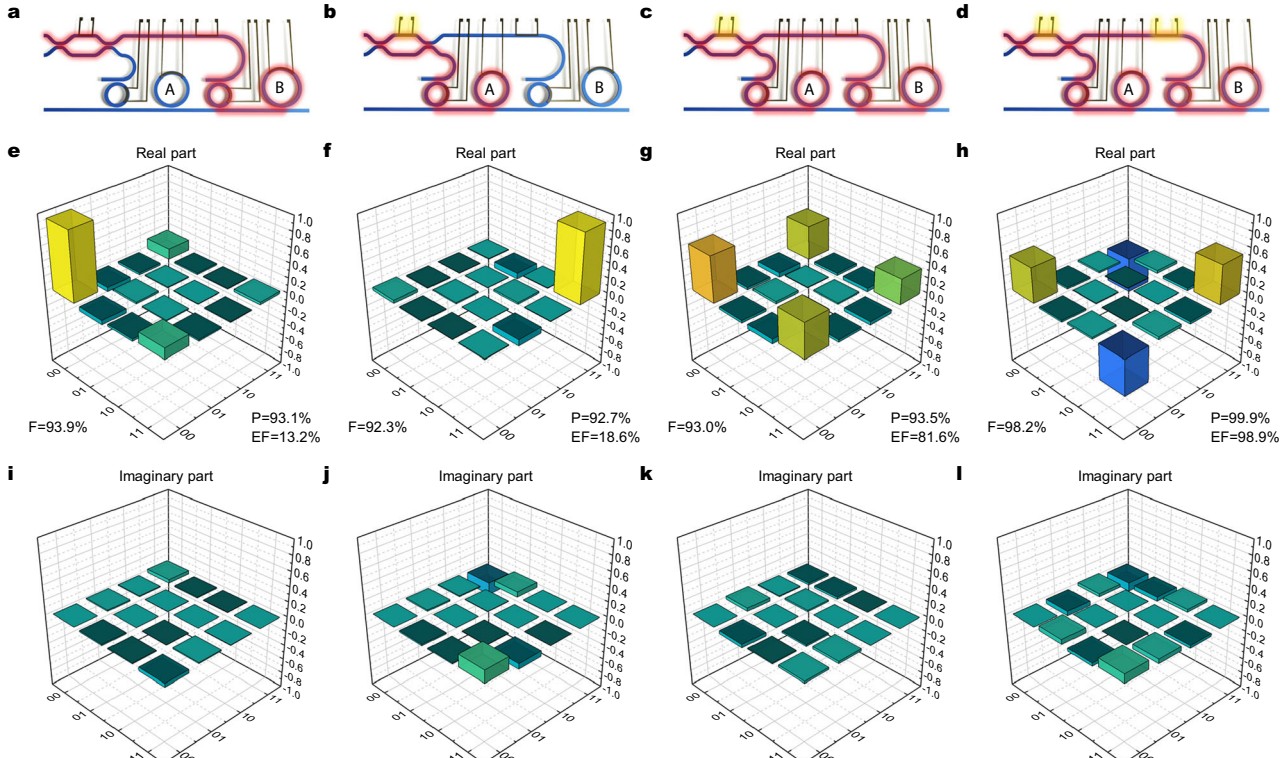

**Fig. 5 | Quantum state tomography in the $\{|00\rangle, |11\rangle\}$ basis (Φ configuration).** Columns from left to right refer respectively to states: $|00\rangle$, $|11\rangle$, $|\Phi^+\rangle$, and $|\Phi^-\rangle$. **a–d** Device pumping scheme for each of the generated state. The path covered by the pump laser is highlighted in red. The generation rings A and B are selectively addressed by acting on the tunable MZI, while the relative phase of the pump is varied through a thermal phase shifter. **e–h** Real and **g–l** imaginary part of the reconstructed density matrices for each of the generated states, estimated through the maximum-likelihood method. $F$, $P$, and $EF$ indicate, respectively, fidelity, purity, and entanglement of formation of each reconstructed state.

sidebands, with a modulation efficiency of around −4.8 dB, corresponding to a modulation index $\beta \approx 1.7$. These losses can be reduced by integrating the modulators on chip. Furthermore, our approach allows the use of frequency-bin spacings potentially much lower than the frequency cutoff of the modulators. This will allow the use of complex wavelength shifting modulation techniques[17,18] to avoid the generation of double sidebands and the consequent 3 dB in added losses.

The resulting spectrum is shown in the lower panels of Fig. 3a and b, in which one can clearly recognize three peaks. Indeed, given the chosen modulated frequency, the central one results from the overlap of the down- and upper-converted original bins. From a quantum optics point of view, this operation achieves quantum interference of the original frequency bins[12] in a similar fashion to what can be done with time bins in a Franson interferometer[19,20]. Here the achievable visibility of quantum interference depends on the correct superposition of the spectra of the modes encoding the two frequency bins for the signal and idler photons, respectively, as outlined in Fig. 4a.

For coincidence counting, the modulated signal and idler photons were filtered using narrowband fiber Bragg gratings to select only the central line at the output of the corresponding modulator, and routed to the single-photon detectors. The results of this experiment are shown in Fig. 4b and c as a function of the modulation frequency. The rapid oscillation of the correlation is due to the different phase acquired by the photons during their propagation from the device to the EOMs. If the resonances share the same $Q$ factor and coupling efficiency, the coincidence rate is proportional to the cross-correlation function (see Supplementary Note 3):

$$G_{s,i}^{(2)}(f_m) = 1 + \frac{\Gamma^2}{(f_m - \Delta/2)^2 + \Gamma^2}\cos\left(4\pi(f_m - \Delta/2)\delta T + 2\varphi_s - 2\varphi_i - \theta\right),$$

(2)

where $\delta T = t_i - t_s$ is the difference between the idler and signal arrival times at the EOMs, and $\varphi_{s(i)}$ is the signal (idler) modulator driving phase. Figure 4b shows good agreement between the experimental results and curve described by Eq. (2) for $\varphi_s - \varphi_i = \theta/2$ and $\delta T = 8.5$ ns, which corresponds to the ~2 m path difference between the idler and signal EOMs in our setup. The curve visibility obtained from a least-square fit of the model is $V = 98.7 \pm 1.2\%$. The two-photon correlation reaches its maximum value $G_{s,i}^{(2)}(f_m) \approx 2$ when $f_m = \Delta/2$, as shown in other works on frequency-bin entanglement[12]. Thanks to the high brightness of the source, coincidence counts on the detectors remain well above the noise level even with the added losses from the modulators, with a CAR level > 50 and detected coincidence rate > 2 kHz, thus implying an interference pattern with a high visibility.

With these results in hand, we set $f_m = \Delta/2$ and varied $\varphi_s$ to perform a Bell-like experiment. The corresponding quantum interference curves are reported in Supplementary Note 2.

## Quantum state tomography

Finally, we show that our device can be operated to generate, directly on chip, frequency-bin photon pairs with a controllable output state. For each of the explored configurations we performed quantum state tomography[21]. First we kept the device in configuration Φ, in which

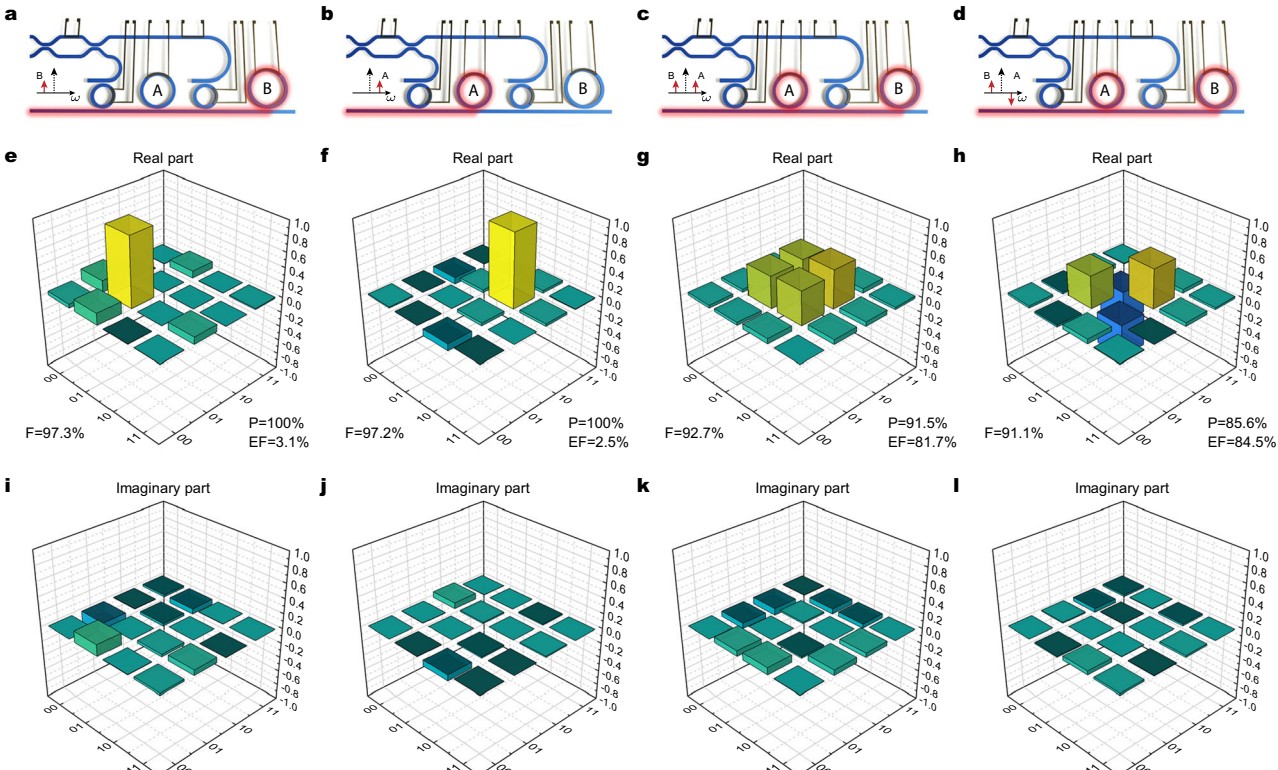

**Fig. 6 | Quantum state tomography in the $\{|01\rangle, |10\rangle\}$ basis ($\Psi$ configuration).** Columns from left to right refer respectively to states: $|01\rangle$, $|10\rangle$, $|\Psi^+\rangle$, and $|\Psi^-\rangle$. **a–d** Device pumping scheme. The bus waveguide is used as input for the pump, whereas the generation rings' resonances are addressed by spectral shaping (modulation) of the pump, performed before coupling to the chip. The relative generation phase between rings A and B is tuned by adjusting the phase of the input modulator driver. **e–l** Reconstructed density matrices for each of the generated states (see caption of Fig. 5 for details).

Ring A and Ring B generate photon pairs in the state $|0\rangle_{s,i}$ and $|1\rangle_{s,i}$, respectively. Thus, the two states of the computational basis $|00\rangle = |0\rangle_s|0\rangle_i$ and $|11\rangle = |1\rangle_s|1\rangle_i$ can be generated by selectively pumping only the appropriate resonator, as shown in Fig. 5a and b. The states were characterized via quantum state tomography[12,21,22], as detailed in the Methods section. In both cases the states are accurately reproduced, with fidelity and purity exceeding 90%.

In a second experiment, the MZI was operated to split the pump power so that the probabilities of generating a photon pair in Ring A and in Ring B are equal. If the pump power is sufficiently low that the probability of emitting two photon pairs is negligible, then the generated frequency bins are in the state $|\Phi(\theta)\rangle$ described by Eq. (1), where the phase factor $\theta$ is controlled by the phase shifter after the MZI. By setting $\theta = 0$ or $\pi$, we were able to generate the two Bell states $|\Phi^+\rangle$ and $|\Phi^-\rangle$, respectively (see Fig. 5c and d). The real and imaginary parts of the density matrix are shown in Fig. 5g, h, k, and l. As expected, we found non-zero off-diagonal terms in the real part of the density matrix, which indicate entanglement. In these cases as well the device is capable of outputting the desired state with purity and fidelity exceeding 90%. The entanglement of formation, a figure of merit to quantify the entanglement of the generated pairs[23], was extracted from the measured density matrices, yielding values > 80% for the two Bell states, in contrast with values < 20% for the two separable states $|00\rangle$ and $|11\rangle$.

Our device can also operate in the $\Psi$ configuration, with the ring resonances arranged as shown in Fig. 1e–g. In this case one is able to generate also the two remaining computational basis states $|01\rangle$, $|10\rangle$ and the two remaining Bell states $|\Psi^+\rangle$ and $|\Psi^-\rangle$. Note that in this configuration, the pump resonances for the two ring resonators are not aligned (Fig. 1f).

When generating the two separable states, either Ring A (to generate $|01\rangle$) or Ring B (to generate $|10\rangle$) was pumped through the bus waveguide by simply tuning the pump to the corresponding resonance (see Fig. 6a and b). To generate the two Bell states, the pump pulse spectrum (which is tuned to be in the middle of the two resonances) is shaped using an external EOM operated at the frequency corresponding to half the difference between the two pump resonances ($f_{m,p} = \Delta_p/2 = 19$ GHz) (see Fig. 6c and d and the Methods section). The pumping ratio and the phase between the two rings were adjusted by tailoring the modulation to obtain an equal probability amplitude of generating a single-photon pair for the states $|01\rangle$ and $|10\rangle$ respectively, while still keeping the probability of double pair generation negligible. The relative phase of the superposition can be controlled by adjusting the EOM driving phase to select either $|\Psi^+\rangle$ or $|\Psi^-\rangle$.

The four generated states were characterized via quantum state tomography as in the previous case. However, we stress that here two different values of bin spacing for the signal ($\Delta_s = 19$ GHz) and idler ($\Delta_i = 3\Delta_s = 57$ GHz) qubits were used. While this does not constitute a problem for the generation of entanglement, as the Hilbert space of the two qubits is built from the tensor product of Hilbert spaces of two qubits with different values for $\Delta_s$ and $\Delta_i$, it offered us the opportunity to demonstrate, for the first time, frequency-bin tomography for uneven spacing. This is done by operating the signal and idler EOMs (see Supplementary Fig. 1) at different frequencies equal to half the frequency spacing of the corresponding resonances.

The experimental results are shown in Fig. 6e–l. All four states were prepared with fidelity close to or exceeding 90%, and purity between 85 and 100%. The entanglement of formation is below 5% for the separable states $|01\rangle$ and $|10\rangle$, while above 80% for the Bell states $|\Psi^+\rangle$ and $|\Psi^-\rangle$, as expected. The reconstructed density matrices show

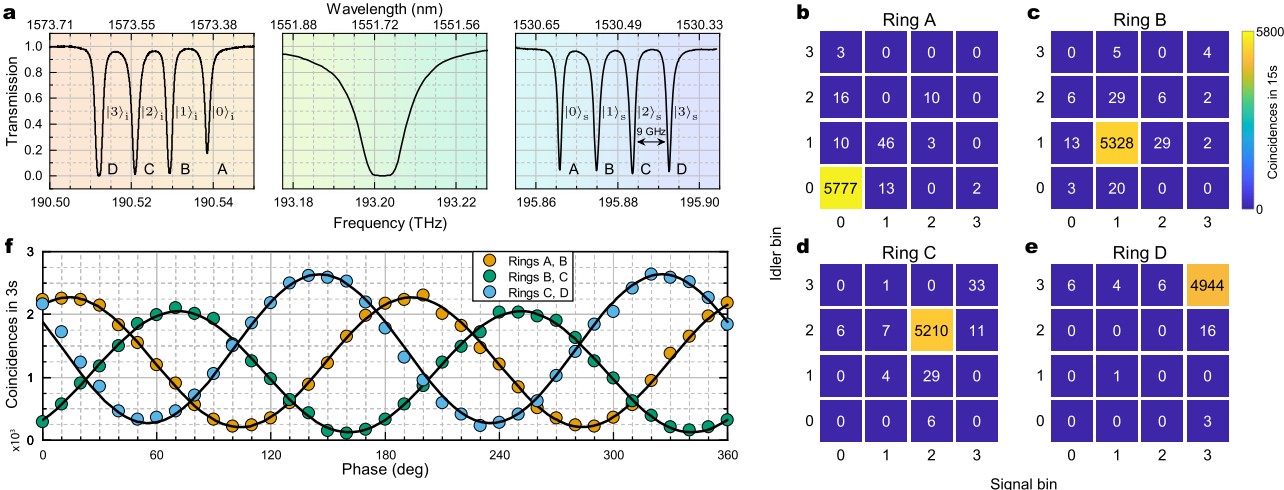

**Fig. 7 | Higher-dimensional states (*qudits*). a** Normalized transmission spectrum of the device used for the generation of higher-dimensional states. The device layout is analogous to the one shown in Fig. 1a, but four generation rings (labeled A, B, C, D) are involved. Panels from left to right show respectively the idler, pump, and signal resonances associated with the correspondent four rings involved. **b**–**e** Correlation matrices showing coincidence counts for each pair of resonators while pumping respectively rings A, B, C, D. **f** Bell-type quantum interference measurements performed on the generated states $|\Phi^+\rangle_{0,1}$ (orange dots), $|\Phi^+\rangle_{1,2}$ (green dots), and $|\Phi^+\rangle_{2,3}$ (blue dots).

increased noise with respect to those reported in Fig. 5, because the modulation efficiency of our idler modulator was significantly reduced at such a high frequency, resulting in additional losses lowering the count rate on the detectors (see the Methods section).

## Scalability to higher-dimensional states

Our approach can be generalized to frequency-bin qudits by scaling the number of coherently excited rings. We give a proof of principle demonstration of this capability by using a different device hosting $d = 4$ rings and add-drop filters. The four sources, labeled A, B, C, D, have radii $R_j = R_0 + j\delta R$ (with $j = 0, ..., d-1$), where $R_0 = 30\,\mu m$ and $\delta R = 0.1\,\mu m$, which leads to a bin spacing of ~9 GHz at 7 FSR from the pump. The spectral response of the device at the output of the bus waveguide, indicated in Fig. 7a, shows the four equidistant bins (labeled 0, 1, 2, 3) associated with the signal and with the idler photons, and the overlapping resonances of the rings at the pump frequency. As in the case of qubits, we used an MZI tree to split the pump into four paths, each feeding a different add-drop ring filter that is used to control the field intensity at the photon pair sources. We focused on the capability to generate the four computational basis states and the two-dimensional Bell states formed by adjacent frequency bins pairs. First, the add-drop filters are tuned on resonance one at a time. This selects the computational basis state that is generated. We characterized those states by performing a *Z*-basis correlation measurement, i.e., by projecting the signal and the idler photon on the *Z*-basis $|l\rangle_s|m\rangle_i$, $l(m) = 0, 1, 2, 3$, in order to measure the uniformity and the crosstalk between the four frequency bins. From the correlation matrices, shown in Fig. 7b–e, it was possible to measure the ratio of the coincidence counts $n_{ll}$ in the frequency-correlated basis $|l\rangle_s|l\rangle_i$ to that in the uncorrelated basis $\sum_{l\neq m} n_{lm}$, and it is about two orders of magnitude. We could compensate for the slightly different amplitude of the different basis states by acting on the MZI tree at the input. Second, the add-drop filters associated with the adjacent frequency-bin-pairs 0–1, 1–2, and 2–3 are tuned on resonance one at a time, thus generating the Bell states $|\Phi^+\rangle_{0,1}$, $|\Phi^+\rangle_{1,2}$ and $|\Phi^+\rangle_{2,3}$, being $|\Phi^+\rangle_{l,m} = (|ll\rangle + |mm\rangle)/\sqrt{2}$. The visibility of quantum interference is assessed by mixing the corresponding frequency bins with the electro-optic modulator. Unlike in the qubit experiment, here we choose a modulation frequency that matches the spectral separation between the bins. We used phase modulators

configured to create first-order sidebands of amplitude equal to that of the baseband, and recorded the coincidences in signal/idler bins 0, 1, 2, and 3. The resulting Bell curves, shown in Fig. 7f, have visibilities $V_{0,1} = 0.831(5)$, $V_{1,2} = 0.884(6)$, and $V_{2,3} = 0.81(1)$, indicating the presence of entanglement between the bin-pairs in all cases. It is worth noting that, as in the two-dimensional case, the relative phase between the three Bell curves in Fig. 7f could be adjusted using on-chip phase shifters in order to realize maximally entangled high-dimensional Bell states.

## Discussion

We demonstrated that a rich variety of separable and maximally entangled states, including any linear superposition of $\{|00\rangle, |11\rangle\}$ or $\{|01\rangle, |10\rangle\}$, can be generated using frequency-bin encoding in a single programmable nano-photonic device, fabricated with existing silicon photonic technologies compatible with multi-project wafer runs. This guarantees that these devices can be available for widespread use in applications, ranging from quantum communication to quantum computing.

Our approach constitutes an innovative paradigm for the integration of frequency-bin devices that goes well-beyond a miniaturization of bulk strategies. Indeed, unlike previous implementations, the states are all generated inside the device, without relying on off-chip manipulation of a single initial state. Controllability of the generated state was shown to be readily accessible on-chip, via electrical control of thermo-optic actuators in one configuration ($\Phi$), and by tailoring the pump spectral properties in another ($\Psi$). In a future version of the device the use of more than two rings for the definition of the state will allow the two configurations to have the same frequency spacing for the qubits. As a result, the device will be capable of generating all four Bell states with the same physical characteristics, as recently demonstrated using an external periodically poled lithium niobate crystal[24]; it will also be used to explore more of the Hilbert space of the two qubits.

Since in our approach the frequency-bin spacing is only limited by the resonator linewidth, the requirements for the electro-optic modulators are greatly relaxed with respect to previous implementations. Indeed, as demonstrated in this work, the frequency-bin separation is compatible with existing silicon integrated modulators[25]. Thus, one can foresee a future evolution of our device that will involve

modulators integrated on-chip. This will further increase its suitability for practical applications, such as quantum key distribution and quantum communications in general. In addition, the ability to independently choose the bin spacing Δ for both qubits, as shown in Fig. 1b–g, demonstrates an additional flexibility in choosing the basis for frequency-bin encoding that can be exploited for the engineering of the source.

The approach demonstrated here is scalable, for one can design and implement devices with more than two generating rings by taking advantage of silicon dense integration, opening the possibility of using frequency qudits instead of simple qubits. As demonstrated in a number of theoretical proposals, such an ability will be of pivotal importance for multiple applications in quantum communication, sensing, and computing algorithms[26]. In addition, our approach could be extended to take advantage of recent progress in all-optical frequency conversion[27,28] to expand the manipulation bandwidth of the frequency bins, thus allowing one to increase the dimension of the accessible Hilbert space enormously.

Finally, our approach allowed us to overcome the trade-off between the frequency-bin spacing and the generation rate that characterized previous work. This was instrumental in achieving a comprehensive assessment of the properties of the generated states, which could be performed using only telecom-grade fiber components—with the sole exception of single-photon detection—with an overall low loss (<4 dB) ensured by the all-fiber technology. The accuracy and the precision that have been achieved in our measurements are state-of-the-art for frequency-bin encoding, even considering results obtained with bulk sources. well-beyond any other reported so far on frequency-bin encoding. All these results will usher in the use of frequency-bin qubits as a practical choice for photonic qubits, capable of combining easy manipulation and robustness for long-haul transmission.

## Methods

### Sample fabrication
The device was fabricated at CEA-Leti (Grenoble), on a 200 mm Silicon-on-Insulator (SOI) substrate with a 220 nm thick top device layer of crystalline silicon on 2 μm thick $SiO_2$ buried oxide. The patterning process of the silicon photonics devices and circuits combines deep ultraviolet (DUV) lithography with 120 nm resolution, inductively coupled plasma etching (realized in collaboration with LTM—Laboratoire des Technologies de la Microélectronique) and $O_2$ plasma resist stripping. Hydrogen annealing was performed in order to strongly reduce etching-induced waveguide sidewalls roughness[29]. After high-density plasma, low-temperature oxide (HDP-LTO) encapsulation—resulting in a 1125 nm thick $SiO_2$ layer—110 nm of titanium nitride (TiN) were deposited and patterned to create the thermal phase shifters, while an aluminium copper layer (AlCu) was used for the electrical pad definition. Finally, a deep etch combining two different steps—$C_4F_8$/$O_2$/CO/Ar plasma running through the whole thickness of both silica upper cladding and buried oxide, followed by a Bosch deep reactive ion etching (DRIE) step to remove 150 μm of the 725 μm thick Si substrate—was implemented to separate the sub-dices, thus ensuring high-quality optical-grade lateral facets for chip-to-fiber edge coupling.

### Linear spectroscopy
The experimental apparatus is schematically represented in Supplementary Fig. 1. The linear characterization of the sample shown in Fig. 1 was realized by scanning the wavelength of a tunable laser (Santec TSL-710), with its polarization controlled by a fiber polarization controller (PC). Light was coupled to the sample at the input of the bus waveguide and collected at the output using a pair of lensed fibers (nominal mode field diameter: 3 μm), with an insertion loss lower than 3 dB/facet. The output signal was detected by an amplified InGaAs photodiode and recorded in real time by an oscilloscope. The resonance

configuration was adjusted by addressing each ring resonator's phase shifter with electric probes driven by multi-channel power supply.

### Nonlinear characterization
The SFWM efficiency for each resonator was assessed through power-scaling experiments (Fig. 2). The flux of generated idler and signal photons was measured by varying the pump power coupled to each microring, while keeping the resonances in place by acting on the thermo-electric phase shifters. The tunable laser source spectrum was filtered by a bandpass (BP) filter in order to reduce the amount of spurious photons at signal and idler frequencies coming from the launching part of the setup, mainly associated with amplified spontaneous emission of the laser diode and Raman fluorescence from the fibers. The collected signal and idler photons were first separated using a coarse wavelength division multiplexer (CWDM), with 2.5 THz (20 nm) nominal channel separation and measured inter-channel crosstalk < −80 dB. The frequency bins of interest were then narrow-band filtered (3 dB-bandwidth: 8 GHz) by a pair of tunable fiber Bragg gratings (FBG): besides selecting the frequency bins with high accuracy, this procedure also suppresses any spurious broadband photon falling outside the bandwidth of the input bandpass filter and not eliminated by the CWDM. The resulting signal and idler photons were routed, using circulators, towards two superconductive single-photon detectors (SSPDs), where time-correlated single-photon counting (TCSPC) was performed with a precision of about 35 ps, mainly determined by the detectors jitter. A coincidence window of $\tau_c = 380$ ps was chosen by selecting the average full width at half maximum (FWHM) of the histogram peak. Accidental counts were estimated from the background level; note that this value is not subtracted from the number of coincidences counted, but was used only to estimate the coincidence-to-accidental ratio, according to the formula:

$$CAR = \frac{\text{total counts in coinc. window} - \text{accidental counts in coinc. window}}{\text{accidental counts in coincidence window}}. \tag{3}$$

### Quantum state tomography
Two-photon interferometry and tomography of the generated quantum states were performed by including a pair of intensity EOMs (iXblue MX-LN) at the signal and idler demultiplexer outputs, coherently driven by a multi-channel RF generator (AnaPico APMS20G). The sidebands of interest were selected by tuning the central stop-band wavelength of the FBGs. The tomography of each quantum state involved 16 individual measurements, each performed in an acquisition time of 15 s. For each measurement, each FBG was tuned to one of the three sideband frequencies obtained from the modulation of the signal (idler) bins, and the EOM's relative phase was adjusted appropriately. Estimation of the density matrices was performed via maximum-likelihood technique[21,22]. For the generation of states in the {|01⟩, |10⟩} basis (Ψ configuration), we added a phase EOM at the input of the setup, coherently driven by the same RF source used for tomography, and we entered the chip at the bus waveguide. The two generation rings were then pumped by the first-order sidebands, while their relative phase was fixed by the phase of the modulation.

### Measurement of qudits
For the Z-basis correlation measurement, a total set of different projectors (for each photon) is used for each basis state. The projector $|l\rangle_s|m\rangle_i$ is implemented by setting the signal(idler) FBG to reflect only the frequency-bin $l(m)$. For those combinations carrying negligible counts (corresponding to frequency uncorrelated bins), the central frequency of the two FBGs cannot be determined by simply maximizing the coincidence rate or the flux of singles in each bin. To

circumvent this, we coupled a secondary laser beam in the counter-propagating direction with respect to that of the pump, and recorded the back reflected light from the sample. The spectra of the latter are monitored after being transmitted by the FBGs, and simultaneously reveal the spectral location of the stop-band of the FBG and the four resonance frequencies of the rings. In this way, the stop-band can be overlapped with the desired frequency bin with high precision.

## Data availability

The data presented in this study are available at https://doi.org/10.5281/zenodo.7464081. Additional data are available from the corresponding authors upon request.

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

## Acknowledgements

This work has been supported by Ministero dell'Istruzione, dell'Università e della Ricerca (Dipartimenti di Eccellenza Program (2018–2022) - F11I18000680001). The device has been designed using the open source Nazca design™ framework.

## Author contributions

M.L. and J.E.S. conceived the original idea. F.A.S., M.L., M.G., and D.B. conceived the the device design. F.A.S and H.E.D. engineered and fabricated the experimental device, under the supervision of C.S. L.Y., C.P., E.P., and C.S. contributed to the engineering and supervised the fabrication. M.C., F.A.S., M.B., and N.T. performed the experimental measurements and data analysis. M.G. and D.B. supervised the the experiments. M.C., F.A.S., M.L., and D.B. developed the theory. N.B. contributed to the analysis of quantum state tomography results. J.E.S. and M.L. supervised the theoretical aspects. M.C., F.A.S., M.B., L.G., J.E.S., M.L., M.G., and D.B. wrote and revised the manuscript. M.L., C.S., M.G., and D.B. coordinated and supervised the project. All authors commented on the manuscript.

## Competing interests

The authors declare no competing interests.
