## [Peer Review File · Nature Communications]

Programmable frequency-bin quantum states in a nano-engineered silicon deviceREVIEWER COMMENTS

Reviewer #1 (Remarks to the Author):

This paper demonstrates the production of two-qubit frequency-bin states using cascaded microring resonators with different free-spectral ranges (FSRs). Modeled after the theoretical proposal in Ref. [16], this approach is able to combine the efficiency available from a large FSR with the ease of manipulation brought about by tighter frequency-bin spacings. The manuscript is well written with very clear and compelling results. I am particularly impressed by the demonstration of "Psi" Bell states which goes beyond the situations considered in [16]. The source design is a major advance forward for frequency-based quantum information, and I am happy to recommend it for publication in Nature Communications.

I have just a few small comments that should be addressed. The only drawback in the results I can see is that Psi states actually occupy a different Hilbert space than the Phi states, since the idler frequency bins possess a different separation. The authors do discuss this point, but some of the statements in the introduction and conclusion seem to give the wrong impression. For example, when it is stated that all four Bell states can be prepared, the basis states are actually different for the two classes of states. So it is not quite true that "any linear superposition" can be prepared, because doing so in fact requires a redefinition of the basis states.

Related to this comment, the authors should mention and relate their setup to recent work which demonstrated all four frequency-bin Bell states using multiline pumping of a PPLN waveguide [arXiv:2205.06141 (2022)]. While I find that setup considerably less elegant than the integrated version here, that version does allow all Bell states to be prepared in the same four frequency bins, compared to the different bins for Phi and Psi here. The authors should note this, as well as elaborate on whether it is possible on their platform to realize all four Bell states in the same bins.

Finally, I think the paper is slightly too strong on the claims about the impressiveness of the CHSH violation and in stating that the "accuracy and precision...are well-beyond any other reported in frequency-bin encoding." There have been several examples with extremely high-fidelity frequency-bin gates [PRL 120, 030502 (2018); PRL 125, 120503 (2020)], and higher-fidelity Bell states have been measured (e.g., the arXiv mentioned above). So these claims should be tempered or removed.

Finally, a couple small clarifications. (1) FGB in line 409 should be FBG. (2) In the text, it seems that the EOM on the pump is a phase modulator, while those used for measurements of the signal and idler are intensity modulators. Is that correct? If so, I would recommend making it clear that the devices in Fig. S1 are different.

Reviewer #2 (Remarks to the Author):

In the manuscript by M. Clementi et al., the authors claim the demonstration of a silicon nanophotonic chip capable of generating frequency entangled two-photon qubits that are directly produced from the photon source, that is, without the use of electro-optical modulators and/or wave shapers outside the source. The photonic chip comprises two microring resonators (A and B) in all-pass configurations and with different free spectral ranges (FSRs, 377 GHz and 373.4 GHz, respectively), as well as a tunable Mach-Zehnder interferometer to control relative phase between the rings A and B. The authors make use of such a platform to demonstrate the four computational qubit basis states (that is, $|00\rangle$, $|01\rangle$, $|10\rangle$, and $|11\rangle$) and the four maximally entangled qubit Bell states ($1/\sqrt{2}(|00\rangle \pm |11\rangle)$ and $1/\sqrt{2}(|10\rangle \pm |01\rangle)$). These are validated by means of quantum interference, CHSH inequality violation, and quantum state tomography measurements. The good quality of the nanophotonic chip quantum source is validated instead by estimating brightness, fidelity, and purity, all of them resulting in a high value (they estimate fidelities close or exceeding 90% and purity between 85% and 100%).

The computational basis states are realized by exciting only one microring, that is, microring A to

obtain $|00\rangle$ and $|10\rangle$ (with phase and no phase modulation, respectively), and microring B to obtain $|11\rangle$ and $|01\rangle$ (with phase and no phase modulation, respectively). The Bell states are obtained instead by pumping both microrings with different phase modulations and Mach-Zehnder interferometer configurations. For Bell state generation, the authors excite a single resonance per microring, in such a way to associate the frequency bins $|0\rangle_s$ and $|0\rangle_i$ to microring B, and $|1\rangle_s$ and $|1\rangle_i$ to microring A. The frequency-bin entanglement generation stems from the uncertainty in the frequency bin associated with each microring resonance.

The integrated platform utilized by the authors is fully compatible with telecom technologies and electro-optic modulation techniques, as the spacing between the frequency bins is limited by the linewidth of the microrings. This implies many applications of the platform, especially in terms of quantum communications.

The work from M. Clementi et al. is absolutely of interest in terms of innovation in integrated nanophotonics both at the scientific and at the engineering level, as well as it is very promising for applications in the field. However, I have some concerns to be clarified and to be convinced of, before recommending the manuscript to be considered for publication in Nature Communications.

The authors claim the scalability of their platform. From my understanding, the number of entangled frequency bins is related to the number of rings, as well as to the number of resonances that are selected for the two rings (in that case, two levels – qubit – as the number of rings and as the number of resonances – one per ring). Is my understanding correct? In that case, I am wondering how this platform is scalable to a higher number of levels by keeping a relatively small footprint. Many works (even not that recent anymore) have demonstrated frequency bin entangled qudits (<https://arxiv.org/pdf/2108.04124.pdf>, Optics Express 26(2), 1825 (2018), Nature 546, 622-626 (2017)). I understand that most of the novelty of this work stems from the Bell state generation directly from the nanophotonic chip, but I am not convinced that limiting the demonstration to qubits is enough for publication in Nature Communications. The rings used in this work show a high FSR (377 GHz and 373.4 GHz). On one hand, such a FSR keeps a small footprint of the microrings. On the other hand, it hampers going to a higher number of frequency modes and then of frequency qudits. Given that, I am wondering what hampers the authors to consider two resonances per ring and get a higher number of frequency modes/levels. Is there any technical issue that prevented the authors to accomplish this in the laboratory? Are the authors able to perform measurements in the laboratory to demonstrate qudits? If qudit Bell states are difficult to reach, at least the computational qudit basis could be demonstrated. Extending this work to the qudit level would increase its impact, as now the demonstration of qubits is becoming obsolete somehow.

I would suggest the authors considering these concerns to improve their work unless they properly address these concerns. If the authors cannot reproduce any experiment (even the most basic, that is computational qudit basis) with the use of qudits and if they do not convince me of the contrary, I cannot recommend the manuscript “Programmable frequency-bin quantum states in a nano-engineered silicon device” by M. Clementi et al. eligible for publication in Nature Communications. Rather, it could be submitted to a more specialized journal.

Reviewer #3 (Remarks to the Author):

The current paper creates two frequency-encoded qubits from silicon ring resonator pair sources on chip, and then electro-optically modulates them off chip to produce quantum interference. The relative phase of the two sources is controlled on-chip to tune between different Bell states. The important claim of the paper is the integration of two photon pair sources and the distribution of the input pump power between them, as well as phase shifter control of the state coming out of the device encoded in frequency. All of this is achieved in a foundry-compatible silicon photonics fabrication run. In fact, I believe the generation of Bell states is a good achievement of the paper. While the results bear some resemblance with Ref 11 and 13, the use of two rings on chip and their relative power split and phase control moves the technology closer to full integration. Compared to Ref 11 which used pulse shapers

(lossy and hard to integrate on chip), the current paper uses EOMs, which are currently off chip but have been integrated on chip in other works, both in silicon and other materials. However, I have concerns with regard to its advance over Ref 11, 13 and other references such as Silverstone et al. Nature Photonics 8, 104 (2014) that prevent me recommending the paper for publication in Nature Communications. I explain my concerns below.

The paper is missing several references to previous work on frequency domain quantum interference, some of which were programmable as well.

- Qing Li et al. Phys. Rev. Applied 12, 054054, 2019

In this paper, Li and colleagues showed both pair generation and subsequent frequency domain manipulation using integrated photonic chips.

- Kobayashi, T. et al. Frequency-domain Hong–Ou–Mandel interference. Nature Photonics 10, 441–444 (2016)

- Joshi et al. Phys. Rev. Lett 124 143601 (2020).

In these papers, quantum interference in the frequency domain over much greater bandwidths was shown.

The paper says, "In frequency-bin encoding today, the generation of photon pairs occurs via spontaneous four-wave mixing in a single ring resonator, with the desired state obtained outside the chip, by means of electro-optical modulators and/or pulse shapers. And since commercial modulators have limited bandwidth, the frequency span separating the photons cannot exceed a few tens of gigahertz, which sets a limit to the maximum free spectral range of the resonator."

This is true for electro-optic modulation but frequency domain manipulation can also occur through nonlinear optical means, as shown by Q. Li et al. in the above 2019 paper. Through nonlinear optical means, the bandwidth is not a limitation and high FSR resonators can easily be used, potentially in an integrated fashion.

Results in Figure 4:

The manner in which the two frequency bins are interfered naturally results in leakage into additional frequency bins, as was also the case in Ref. 11. This kind of leakage is absent in spatial HOM experiments. In fact, Ref. 11 was not the first to show quantum interference between different frequency bins - it was shown much earlier by Kobayashi et al., and there the leakage into additional bins was absent. Could the authors comment on how this kind of leakage could be avoided, and what are the deleterious effects on the performance of the device from a quantum information processing point of view?

In Fig. 4, I would recommend the authors to plot raw counts instead of normalized counts, as it is not clear if turning the modulation ON results in a reduction in the count rate.

Additionally, the electro-optic modulators off chip are lossy. Since the conversion efficiency of the DSB-SC modulation is -4.8 dB, and there would be additional insertion loss, could the authors comment if the CAR was reduced when frequency domain quantum interference was performed, compared to the 100:1 CAR that is reported in Fig. 2? Could the authors discuss if this is an issue and ways to mitigate it?

Minor comments:

There is a singular mention of biphoton amplitudes in the main text on page 3, and a detailed discussion in the SI. It would help to briefly summarize the discussions in the main text, since this information could be important for interference and state generation.

Is the MZI balanced in the absence of a voltage applied to the top arm of the MZI? If yes, then figure 6a should route the input to the bottom port.

Answer to reviewer 1

This paper demonstrates the production of two-qubit frequency-bin states using cascaded microring resonators with different free-spectral ranges (FSRs). Modeled after the theoretical proposal in Ref. [16], this approach is able to combine the efficiency available from a large FSR with the ease of manipulation brought about by tighter frequency-bin spacings. The manuscript is well written with very clear and compelling results. I am particularly impressed by the demonstration of "Psi" Bell states which goes beyond the situations considered in [16]. The source design is a major advance forward for frequency-based quantum information, and I am happy to recommend it for publication in Nature Communications.

We thank the reviewer for recognizing the importance of our work, and for explicitly recommending it for publication in Nature Communication.

I have just a few small comments that should be addressed. The only drawback in the results I can see is that Psi states actually occupy a different Hilbert space than the Phi states, since the idler frequency bins possess a different separation. The authors do discuss this point, but some of the statements in the introduction and conclusion seem to give the wrong impression. For example, when it is stated that all four Bell states can be prepared, the basis states are actually different for the two classes of states. So it is not quite true that "any linear superposition" can be prepared, because doing so in fact requires a redefinition of the basis states.

We agree with the referee that our introduction could be misunderstood as implying the ability to span the whole Hilbert space. We thank the reviewer for pointing this out. We modified a part of the introduction to avoid any confusion:

“We demonstrate that with the very same device one can generate all superpositions of the $|00\rangle$ and $|11\rangle$ states or, in another configuration with different frequency bin spacing, all superpositions of the $|01\rangle$ and $|10\rangle$ states. One needs only to drive the on-chip phase shifter and set the pump configuration appropriately. This means that all four fully-separable states of the computational basis and all four maximally-entangled Bell states ($|\Phi^\pm\rangle = (|00\rangle \pm |11\rangle)/\sqrt{2}$ and $|\Psi^\pm\rangle = (|01\rangle \pm |10\rangle)/\sqrt{2}$) are accessible.”

Related to this comment, the authors should mention and relate their setup to recent work which demonstrated all four frequency-bin Bell states using multiline pumping of a PPLN waveguide [arXiv:2205.06141 (2022)]. While I find that setup considerably less elegant than the integrated version here, that version does allow all Bell states to be prepared in the same four frequency bins, compared to the different bins for Phi and Psi here. The authors should note this, as well as elaborate on whether it is possible on their platform to realize all four Bell states in the same bins.

We thank the reviewer for pointing out the recent work on PPLN crystals, which is indeed very interesting and relevant for our work. This contribution was added to our

references. To further clarify the point of the frequency bin spacing, we have modified a part of the conclusions, which now reads

“Controllability of the generated state was shown to be readily accessible on-chip, via electrical control of thermo-optic actuators in one configuration (Φ), and by tailoring the pump spectral properties in another (Ψ). In a future version of the device the use of more than two rings for the definition of the state will allow the two configurations to have the same frequency spacing for the qubits. As a result, the device will be capable of generating all four Bell states with the same physical characteristics, as recently demonstrated using an external periodically-poled lithium niobite crystal [arXiv:2205.06141]; it will also be used to explore additional parts of the Hilbert space of the two qubits.”

Finally, I think the paper is slightly too strong on the claims about the impressiveness of the CHSH violation and in stating that the "accuracy and precision...are well-beyond any other reported in frequency-bin encoding." There have been several examples with extremely high-fidelity frequency-bin gates [PRL 120, 030502 (2018); PRL 125, 120503 (2020)], and higher-fidelity Bell states have been measured (e.g., the arXiv mentioned above). So these claims should be tempered or removed.

We agree with the reviewer that this statement on the Bell measurements should be tempered, for it does not take into account some work realized with bulk sources.

The sentence now reads:

...are state-of-the-art for frequency-bin encoding, even considering results obtained with bulk sources."

In addition, the entire section of Bell measurements was moved to Supplementary Information.

Finally, a couple small clarifications. (1) FGB in line 409 should be FBG. (2) In the text, it seems that the EOM on the pump is a phase modulator, while those used for measurements of the signal and idler are intensity modulators. Is that correct? If so, I would recommend making it clear that the devices in Fig. S1 are different.

We thank the reviewer for pointing out these mistakes, which have been corrected in the revised text.

Answer to reviewer 2

In the manuscript by M. Clementi et al., the authors claim the demonstration of a silicon nanophotonic chip capable of generating frequency entangled two-photon qubits that are directly produced from the photon source, that is, without the use of electro-optical modulators and/or wave shapers outside the source. The photonic chip comprises two microring resonators (A and B) in all-pass configurations and with different free spectral ranges (FSRs, 377 GHz and 373.4 GHz, respectively), as well as a tunable Mach-Zehnder interferometer to control relative phase between the rings A and B. The authors make use of such a platform to demonstrate the four computational qubit basis states (that is, $|00\rangle$, $|01\rangle$, $|10\rangle$, and $|11\rangle$) and the four maximally entangled qubit Bell states ($1/\sqrt{2}(|00\rangle \pm |11\rangle)$ and $1/\sqrt{2}(|10\rangle \pm |01\rangle)$). These are validated by means of quantum interference, CHSH inequality violation, and quantum state tomography measurements. The good quality of the nanophotonic chip quantum source is validated instead by estimating brightness, fidelity, and purity, all of them resulting in a high value (they estimate fidelities close or exceeding 90% and purity between 85% and 100%).

The computational basis states are realized by exciting only one microring, that is, microring A to obtain $|00\rangle$ and $|10\rangle$ (with phase and no phase modulation, respectively), and microring B to obtain $|11\rangle$ and $|01\rangle$ (with phase and no phase modulation, respectively). The Bell states are obtained instead by pumping both microrings with different phase modulations and Mach-Zehnder interferometer configurations. For Bell state generation, the authors excite a single resonance per microring, in such a way to associate the frequency bins $|0\rangle_s$ and $|0\rangle_i$ to microring B, and $|1\rangle_s$ and $|1\rangle_i$ to microring A. The frequency-bin entanglement generation stems from the uncertainty in the frequency bin associated with each microring resonance.

The integrated platform utilized by the authors is fully compatible with telecom technologies and electro-optic modulation techniques, as the spacing between the frequency bins is limited by the linewidth of the microrings. This implies many applications of the platform, especially in terms of quantum communications.

The work from M. Clementi et al. is absolutely of interest in terms of innovation in integrated nanophotonics both at the scientific and at the engineering level, as well as it is very promising for applications in the field. However, I have some concerns to be clarified and to be convinced of, before recommending the manuscript to be considered for publication in Nature Communications.

We thank the referee for recognizing the importance of our work, and for clearly stating its impact for applications in quantum technologies.

The authors claim the scalability of their platform. From my understanding, the number of entangled frequency bins is related to the number of rings, as well as to the number or resonances that are selected for the two rings (in that case, two levels – qubit – as the number of rings and as the number of resonances – one per ring). Is my understanding

correct? In that case, I am wondering how this platform is scalable to a higher number of levels by keeping a relatively small footprint.

Many works (even not that recent anymore) have demonstrated frequency bin entangled qudits (<https://arxiv.org/pdf/2108.04124.pdf>, Optics Express 26(2), 1825 (2018), Nature 546, 622-626 (2017)). I understand that most of the novelty of this work stems from the Bell state generation directly from the nanophotonic chip, but I am not convinced that limiting the demonstration to qubits is enough for publication in Nature Communications. The rings used in this work show a high FSR (377 GHz and 373.4 GHz). On one hand, such a FSR keeps a small footprint of the microrings. On the other hand, it hampers going to a higher number of frequency modes and then of frequency qudits.

The footprint of our source was never mentioned in the manuscript, for its design **was not optimized to be compact**. Nonetheless, even without optimization we believe the device size required by our approach is impressively small. Consider, as an example, the case of 10-level qudits, with 20 GHz frequency spacing. When operating in the “Psi” configuration with the rings pumped through the same bus waveguide, our approach would require ten rings with a radius of about 30 μm . Such a structure could be easily accommodated on an area of 0.065 mm^2 . For comparison, a single-ring source with the same frequency spacing should have a radius of about 0.8 mm and would occupy an area ten times larger.

Thus, our approach simply takes advantage of the complexity of structures that can be currently fabricated in silicon photonics. This can easily include several tens or even hundreds of circuitual elements, many more than those necessary to implement our source, even with a large number of qudits.

Given that, I am wondering what hampers the authors to consider two resonances per ring and get a higher number of frequency modes/levels. Is there any technical issue that prevented the authors to accomplish this in the laboratory?

We agree with the reviewer that one could take advantage of more than one resonance per ring at the cost of increasing the ring radius. This would combine the traditional approach with ours, which makes use of multiple rings. We were aware of this point, but we chose to work with small rings and one resonance per ring, as this is the simplest case and the one in which the advantages of our approach are more evident.

The focus of this work is: (a) the design of an integrated device capable of high emission rates (thus a small radius) with frequency bin spacing low enough to be compatible with telecom grade modulators; and (b) programmability of the generated state. The combination of these two features clearly set our work apart from previous literature.

We have added a paragraph to the Supplementary Information where we outline the effective footprint of our device in the cases described in the answer above.

Are the authors able to perform measurements in the laboratory to demonstrate qudits? If qudit Bell states are difficult to reach, at least the computational qudit basis could be

demonstrated. Extending this work to the qudit level would increase its impact, as now the demonstration of qubits is becoming obsolete somehow.

We do not share the reviewer's sense that the demonstration of qubits is "obsolete". The value of such demonstrations for any future work depends very much on the **degree of freedom** that is used, the **platform**, and the **quality of the results**. These three aspects are crucial in any application. In this respect, our integrated programmable source of frequency-bin qubits is unique, and its performance is remarkable. However, we fully agree with the reviewer that qudits are particularly interesting, especially in photonic systems.

The reviewer is requesting additional measurements to demonstrate that qudits are do-able with our platform. She/he suggests we demonstrate qudits using the azimuthal modes of the rings. However, we believe that this would be in contradiction with the core message of the present work, which is the idea that each frequency-bin is associated with a different ring. In addition, the spacing between some of the energy levels would not be compatible with existing telecom equipment. Furthermore, demonstrating the computational basis states, as requested by the reviewer, would require turning off some azimuthal modes while tuning others on. This functionality cannot be achieved using a single ring, so we would need a new device where coupled ring resonators are used to selectively switch off FWM on given resonances. On the other hand, demonstrating qudits using our design requires a device with additional rings coupled to the bus waveguide. In either case a new sample would be needed, and this requires a fabrication run in some open fab service, which typically takes about one year for just the realization of the device.

That said, moving toward the realization of qudits was already among our goals, as was also stated in the conclusion of the manuscript. For this reason, we had the fabrication of a device for the realization of a qudit generation already well under way; we have very recently received this device. We were therefore able to perform preliminary experiments showing that four rings can be operated simultaneously to generate four-level qudits. We show that the individual levels in the qudits can be separately addressed by performing coincidence experiments operating each ring (the condition to generate the two qudit computational basis states $|00\rangle$, $|11\rangle$, $|22\rangle$ and $|33\rangle$). Furthermore, we demonstrate quantum coherence by measuring pairwise Bell curves between the energy levels, while operating two rings at a time.

While a complete study of the various separable and entangled qudit states that can be generated on demand in our device is well-beyond the scope of the present work, these results prove that our design is scalable to the qudit level.

We have added a figure to the paper describing these preliminary results on qudits, and a corresponding paragraph to describe them, as follows:

"Our approach can be generalized to frequency bin qudits by scaling the number of coherently excited rings. We give a proof of principle demonstration of this capability

by using a different device hosting $d = 4$ rings and add-drop filters. The four sources, labelled A, B, C, D , have radii $R_j = R_0 + j\delta R$ (with $j = 0, \dots, d - 1$), where $R_0 = 30 \mu m$ and $\delta R = 0.1 \mu m$, which leads to a bin spacing of approximately $9 GHz$ at $7 FSR$ from the pump. The spectral response of the device at the output of the bus waveguide, indicated in Fig. 7a, shows the four equidistant bins (labeld 0,1,2,3) associated with the signal and with the idler photons, and the overlapping resonances of the rings at the pump frequency. As in the case of qubits, we used a tree of MZI to split the pump into four paths, each feeding a different add-drop ring filter that is used to control the field intensity at the photon pair sources. In our demonstration, we focused on the capability to generate the four computational basis states and the two-dimensional Bell states formed by adjacent frequency bins pairs. First, the add-drop filters are tuned on resonance one at a time. This selects the computational basis state that is generated. We characterized those states by performing a Z-basis correlation measurement, i.e., by projecting the signal and the idler photon on the Z-basis $\{|l\rangle_s |m\rangle_i\}$, $l(m) = 0,1,2,3$, in order to measure the uniformity and the cross-talk between the four frequency bins. From the correlation matrices, shown in Fig.7 b-e, it was possible to measure the ratio between the coincidence counts n_{ll} in the frequency-correlated basis $|l\rangle_s |l\rangle_i$ to that in the uncorrelated basis $\sum_{l \neq m} n_{lm}$, and it is about two orders of magnitude. It was possible to compensate for the slightly different amplitude of the different basis states by acting on the MZI tree at the input. Second, the add-drop filters associated to the adjacent frequency bin pairs $0 - 1$, $1 - 2$ and $2 - 3$ are tuned on resonance one at a time, thus generating the Bell states $|\Phi\rangle_{0,1}$, $|\Phi\rangle_{1,2}$ and $|\Phi\rangle_{2,3}$. The visibility of quantum interference is assessed by mixing the corresponding frequency bins with the electro-optic modulator. Contrary to the two-dimensional case, here we choose a modulation frequency that matches the spectral separation between the bins. We used phase modulators configured to create first order sidebands of amplitude equal to that of the baseband, and recorded the coincidences in signal/idler bins 0,1, 2 and 3. The obtained Bell curves, shown in Fig. 7e, have visibilities $V_{0,1} = 0.831(5)$, $V_{1,2} = 0.884(6)$ and $V_{2,3} = 0.81(1)$, indicating the presence of entanglement between the bin-pairs in all cases. It is worth noting that, as in the two-dimensional case, the relative phase between the three Bell curves in Fig. 7e could be adjusted using on-chip phase shifters in order to realize maximally entangled high-dimensional Bell states”.

We have also added the following paragraph to the “Materials and Methods” section:

“For the Z-basis correlation measurement, a total set of different projectors (for each photon) are used for each basis state. The projector $|l\rangle_s |m\rangle_i$ is implemented by setting the signal(idler) FBG to reflect only the frequency bin $l(m)$. For those combinations carrying negligible counts (corresponding to frequency uncorrelated bins), the central frequency of the two FBGs cannot be determined by simply maximizing the coincidence rate or the flux of singles in each bin. To circumvent this, we coupled a secondary laser beam in the counter-propagating direction with respect to that of the pump, and recorded the back reflected light from the sample. The spectra of the latter are monitored after

being transmitted by the FBGs, and simultaneously reveal the spectral location of the stop band of the FBG and the four resonance frequencies of the rings. In this way, the stop band can be overlapped to the desired frequency bin with high precision.”

We believe that this modification to our work fully meets the reviewer’s request.

I would suggest the authors considering these concerns to improve their work unless they properly address these concerns. If the authors cannot reproduce any experiment (even the most basic, that is computational qudit basis) with the use of qudits and if they do not convince me of the contrary, I cannot recommend the manuscript “Programmable frequency-bin quantum states in a nano-engineered silicon device” by M. Clementi et al. eligible for publication in Nature Communications. Rather, it could be submitted to a more specialized journal.

The experimental demonstration that our design can be efficiently used for the generation of qudits provided in the present version of the manuscript removes any doubt on scalability. Answering this concern was certainly not trivial - and we were lucky to have a sample on the way! - but we thank the reviewer, who certainly stimulated us to achieve an even stronger result.

We are therefore confident that the reviewer will now find our work suitable for publication in Nature Communications.

Answer to reviewer 3

The current paper creates two frequency-encoded qubits from silicon ring resonator pair sources on chip, and then electro-optically modulates them off chip to produce quantum interference. The relative phase of the two sources is controlled on-chip to tune between different Bell states. The important claim of the paper is the integration of two photon pair sources and the distribution of the input pump power between them, as well as phase shifter control of the state coming out of the device encoded in frequency. All of this is achieved in a foundry-compatible silicon photonics fabrication run. In fact, I believe the generation of Bell states is a good achievement of the paper. While the results bear some resemblance with Ref 11 and 13, the use of two rings on chip and their relative power split and phase control moves the technology closer to full integration. Compared to Ref 11 which used pulse shapers (lossy and hard to integrate on chip), the current paper uses EOMs, which are currently off chip but have been integrated on chip in other works, both in silicon and other materials. However, I have concerns with regard to its advance over Ref 11, 13 and other references such as Silverstone et al. Nature Photonics 8, 104 (2014) that prevent me recommending the paper for publication in Nature Communications. I explain my concerns below.

We thank the reviewer for pointing out several elements of novelty of our work with respect to the current literature on frequency-bin entanglement, such as Ref. 11 and 13. However, she/he has some concerns regarding the significance of these advances over those references, as well as other work involving the Hong-Ou-Mandel (HOM) effect. We believe that such concerns might have originated from a lack of sufficient clarity of the original version of the manuscript, which we have now amended. To better illustrate our viewpoint, we address in detail the reviewer's comments.

First, we want to emphasize that, unlike the work in Ref. 11 and 13, the focus of our work is **on-chip programmability** without the need of external devices. In addition, compared to earlier work, where frequency-bins were generated in a single ring resonator, our device is composed by several integrated components that:

1. can be **controlled through on-chip electrical actuators**;
2. are compatible with **silicon** CMOS technology;
3. can **generate different states** depending on the chosen configuration (i.e. " ϕ " vs " Ψ ");
4. **overcome the trade-off between free spectral range and source brightness intrinsic to a single ring**, allowing a regime of operation compatible with commercial (and integrated) electro-optic modulators.

In addition, the new experimental data provided in the revised version of the manuscript demonstrate that **scaling-up to on-chip programmable qudits** is possible using

our design, further setting the present work apart from the previous literature. We believe that these results represent a significant advance compared to the work presented in Refs. 11 and 13, especially from a quantum engineering standpoint.

The reviewer is also concerned about the novelty of our work with respect to that of Silverstone et al., published in Nature Photonics 8, 104 (2014). We point out that there are at least three fundamental differences between their work and ours:

1. Silverstone et al. report on Hong-Ou-Mandel dip experiments, while our work deals with Bell-like experiments. We recognize that this difference might have been misunderstood in the first version of the manuscript, which we have therefore revised for sake of clarity (see response to next comment). To briefly summarize what is discussed later, we emphasize that HOM effect emerges from the quantum interference of identical *photons* (i.e. number states in the Fock basis) at a beamsplitter (or analogous device), while what we address here is the quantum interference between frequency bin *qubits*, which bears a similarity with Franson-type interferometry, rather than HOM effect;
2. Silverstone et al. exploit dual rail (i.e. path-encoded) qubits, while our work uses frequency-bin encoded entangled states;
3. Silverstone et al. deal with non-resonant structures, while our work exploits integrated micro-resonators.

Thus, when compared to our manuscript, the work of Silverstone et al. reported in Nature Photonics 8, 104 (2014) deals with 1) *a different quantum effect* relying on 2) *a different encoding* via 3) *different sources*. Therefore, while we agree that at first glance one could find similarities between our work and that of Silverstone and co-workers, we do not believe that it constitutes a precedent for our results.

We thank the reviewer for bringing to our attention these possible misunderstandings. To further underline the importance of our work with respect to the previous literature, we have added the following sentence in the Introduction:

“These two breakthroughs, namely high emission rates in combination with high values of the free spectral range, together with output state control using on-chip components, are only possible using multiple rings: they would not be feasible were the frequency bins encoded on the azimuthal modes of a single resonator.”

We have also added a citation to Silverstone et al., Nature Photonics 8, 104 (2014), when mentioning dual rail qubits in the introduction.

The paper is missing several references to previous work on frequency domain quantum interference, some of which were programmable as well.

- Qing Li et al. Phys. Rev. Applied 12, 054054, 2019

In this paper, Li and colleagues showed both pair generation and subsequent frequency domain manipulation using integrated photonic chips.

- Kobayashi, T. et al. Frequency-domain Hong–Ou–Mandel interference. Nature Photonics 10, 441–444 (2016)

- Joshi et al. Phys. Rev. Lett 124 143601 (2020).

In these papers, quantum interference in the frequency domain over much greater bandwidths was shown.

We acknowledge that the reviewer’s comment highlights an important possible source of confusion in our text, and we thank the reviewer for giving us the opportunity to address this issue.

We would like to emphasize that we are not performing Mandel-dip (i.e. HOM interference) experiments but, rather, experiments on the violation of Bell inequalities. In Mandel-dip experiments, such as in the three papers cited by the referee (and in the work by Silverstone et al. mentioned in the comment above), two photons need to interact by some form of quantum interference (for instance on a beamsplitter, in the case of path encoding), meaning that they need to be in the same place at the same time for the higher-order interference to take place. In contrast, in the case of Bell-type experiments - such as Franson type experiments, as well as the one presented here - the two photons (here signal and idler) propagate on two completely different paths, never “meeting” each other after separation. There is therefore no possibility for the two photons to produce HOM-type interference after exiting the chip, neither in the spatial, nor in the frequency domain.

The three articles cited above refer specifically to single-photon frequency conversion and two-photon Hong-Ou-Mandel dip experiments, which is not the subject of our manuscript. Instead, we are reporting on the programmable generation of entangled *qubits* (and, in the revised version of the manuscript, also *qudits*). Please note that neither the generation of entangled states, nor the generation of programmable quantum states, nor encoding in frequency bin, are reported in any of the above-cited papers.

The paper says, "In frequency-bin encoding today, the generation of photon pairs occurs via spontaneous four-wave mixing in a single ring resonator, with the desired state obtained outside the chip, by means of electro-optical modulators and/or pulse shapers. And since commercial modulators have limited bandwidth, the frequency span separating the photons cannot exceed a few tens of gigahertz, which sets a limit to the maximum free spectral range of the resonator."

This is true for electro-optic modulation but frequency domain manipulation can also occur through nonlinear optical means, as shown by Q. Li et al. in the above 2019 paper. Through nonlinear optical means, the bandwidth is not a limitation and high FSR resonators can easily be used, potentially in an integrated fashion.

We acknowledge that our experiment the analysis of the generated state involves frequency-conversion after outputting the chip, and therefore it might be worth comparing the methodology employed here (i.e. electro-optic modulation) with other state-of-the-art approaches. Currently, all-optical frequency conversion provides a much wider bandwidth than electro-optic modulation, allowing in principle to overcome the limitations associated to the latter. However, it should be remarked that the former does not have the same maturity and performance level as electro-optic modulation, and requires the use of cumbersome and expensive external set-ups. For instance, in the works suggested by the reviewer:

1. in [Qing Li et al.] external filters are used to separate the photons and a high quality factor ring resonator on a second chip is used for frequency conversion: the high finesse resonances on the second ring resonator must therefore be kept in fine tuning with the generating ring, limiting the bandwidth and the scope of the experiment;
2. Kobayashi, T. et al. use bulk PPLN crystals pumped by powerful lasers to achieve low efficiency light conversion, posing an inherent limitation to integrability;
3. Joshi et al. employ Bragg scattering Four Wave Mixing in a 100 meters long dispersion-shifted fiber to achieve partial frequency overlap. While certainly interesting, this approach involving a 100 m-long fiber certainly cannot be used “in an integrated fashion”.

In contrast, we note that in our work the use of multiple rings, and therefore closely spaced resonances, allows us to greatly reduce the bandwidth required to achieve Bell-type quantum interference compared to the above examples, providing a device capable of integration.

As a final remark, we note that our system could, in principle, be indeed used to produce HOM-type interference. This would imply generating single photon states on both idler bins 0 and 1 respectively (i.e. two distinct pairs, one per ring) and using the signal photons for heralding. After frequency mixing (i.e. modulation) of the signal and idler photons generated, it would be possible, in principle, to show anti-bunching. This would correspond, to a certain extent, to what is done by Kobayashi and coworkers, and by Silverstone and coworkers in the aforementioned works. While this kind of experiment would indeed be interesting, and it would also take advantage of the key-features of our system (reconfigurability, brightness, integrability), it would constitute

the starting point for a completely different work, which is out of the scope of the present manuscript.

We hope this clarification had dispelled any doubt concerning the novelty of the present results with respect to existing work on HOM-type interference. To clarify this point in the manuscript, and to remove any possible source of confusion to the reader, we have modified two sentences when describing Fig. 4a as follows:

“From a quantum optics point of view, this operation achieves quantum interference of the original frequency bins [12] in a fashion similar to what can be done with time bins in a Franson interferometer [20, 21]. Here the achievable visibility of quantum interference depends on the correct superposition of the spectra of the modes encoding the two frequency bins for the signal and idler photons respectively, as outlined in Fig. 4a.”

We thank the reviewer for her/his comment, as we agree that all-optical frequency conversion could offer an important extension of our work. Indeed, having narrowly separated frequency bins, as those achievable with our approach, and wideband frequency conversion, as the one offered by all-optical approaches, will allow one to increase the dimension of the Hilbert space well-beyond what could be done with only EOMs. We add the following sentence to the conclusions:

In addition, our approach could be extended to take advantage of recent progress in all-optical frequency conversion [Kobayashi2016, Li2019] to expand the manipulation bandwidth of the frequency-bins, thus allowing one to increase the dimension of the accessible Hilbert space enormously.

Results in Figure 4:

The manner in which the two frequency bins are interfered naturally results in leakage into additional frequency bins, as was also the case in Ref. 11. This kind of leakage is absent in spatial HOM experiments. In fact, Ref. 11 was not the first to show quantum interference between different frequency bins - it was shown much earlier by Kobayashi et al., and there the leakage into additional bins was absent. Could the authors comment on how this kind of leakage could be avoided, and what are the deleterious effects on the performance of the device from a quantum information processing point of view? In Fig. 4, I would recommend the authors to plot raw counts instead of normalized counts, as it is not clear if turning the modulation ON results in a reduction in the count rate.

The answer to this comment directly follows the previous one. We understand that Fig. 4b in our work may appear, at a first glance, very similar to Fig 3b,e and d in [Silverstone et al., Nature Photonics 8, 104 (2014)]. However, as discussed above, we remark that the curve in our work reports a very different quantum effect. The curve in

our Fig. 4b is not given by direct interference of the two photons on some form of beamsplitter, as in [Kobayashi et al.]. The two photons instead propagate on two separate fibers, and never directly interact after emission, similarly to what is done in Bell-type experiments. What is represented in Fig. 4b of our work is indeed a kind of Bell curve at fixed angle, where the interference is modulated to disappear by making the states increasingly distinguishable due to imperfect wavefunction overlap in the frequency domain (akin to spatially or temporally separating the two polarizations in polarization-encoded entangled photon pairs).

In this sense, the effect of leakage due to modulation is to be considered equivalent to any other loss – as for instance coupling losses, filtering losses or detection losses – present in any Bell experiment reported in literature (with the few exceptions of loophole-free Bell inequality violations). Indeed, the fact that our source has high brightness makes the output quantum state very resistant to losses, as can be clearly seen by the high visibility of the interference. We thank the referee for giving us the opportunity to explicitly point this out by stating the CAR and the total coincidence rate.

We have now added the following to the main text:

“Thanks to the high brightness of the source, coincidence counts on the detectors remain well above the noise level even with the added losses from the modulators, with a CAR level > 50 and detection rate $> 2\text{kHz}$, thus implying an interference pattern with a high visibility.”

Additionally, the electro-optic modulators off chip are lossy. Since the conversion efficiency of the DSB-SC modulation is -4.8 dB , and there would be additional insertion loss, could the authors comment if the CAR was reduced when frequency domain quantum interference was performed, compared to the 100:1 CAR that is reported in Fig. 2? Could the authors discuss if this is an issue and ways to mitigate it?

As discussed in the comment above, losses due to the use of external modulators do not represent, in our experiment, a limiting factor, as both the Bell's curve visibility and the CAR values remain very high.

Nevertheless, we agree with the referee that the reduction of losses in integrated quantum optical experiments is always desirable. And indeed, if one uses our design with integrated modulators, there are techniques at radio frequencies using traveling wave modulation that have been adapted to optical domain for single sideband generation without leakage. Another possible approach to improve the modulation efficiency would be to adopt acousto-optic modulation (and more in general Brillouin effect), which was recently shown to be heterogeneously integrable with silicon photonics technology.

Moreover, our device has the unique property that the generation rate is independent of the frequency-bin spacing, which makes it possible to use bin spacings much lower than the cut-off frequencies of commercially available modulators. This also opens the possibility of using much more complex and efficient frequency conversion methods with respect to driving modulators with simple monotonics.

Here again we thank the referee for the opportunity to highlight this additional advantage provided by our device. We have added the following sentence to the main text, just after stating the losses of the modulators:

“These losses can be reduced by integrating the modulators on chip. Furthermore, our approach allows the use of frequency bin spacings potentially much lower than the frequency cut-off of the modulators. This will allow the use of complex wavelength shifting modulation techniques [M. L. Riazat, G. F. Virshup and J. N. Eckstein, "Optical wavelength shifting by traveling-wave electrooptic modulation," in IEEE Photonics Technology Letters, vol. 5, no. 9, pp. 1002-1005, Sept. 1993, doi: 10.1109/68.257172; Kittlaus, E.A., Jones, W.M., Rakich, P.T. et al, Nat. Photonics 15, 43–52 (2021). Doi:10.1038/s41566-020-00711-9] to avoid the generation of double sidebands and the consequent 3 dB in added losses.”

Minor comments:

There is a singular mention of biphoton amplitudes in the main text on page 3, and a detailed discussion in the SI. It would help to briefly summarize the discussions in the main text, since this information could be important for interference and state generation.

We believe that a detailed discussion on the bi-photon wavefunction would hinder the readability of the manuscript. To help the reader, we have added a footnote reference to the relevant section of the Supplementary Information when we mention the bi-photon wavefunction in the main text.

Is the MZI balanced in the absence of a voltage applied to the top arm of the MZI? If yes, then figure 6a should route the input to the bottom port.

We thank the referee for pointing out this possible source of confusion; we have added the following sentence to the Supplementary Information:

“The Mach Zehnder interferometer was designed so that, with no bias applied, the pump is routed to the output on the top as represented in Fig. 5a.”

REVIEWERS' COMMENTS

Reviewer #2 (Remarks to the Author):

The authors have successfully addressed all the comments and concerns that I have made in order to consider their work suitable for publication. The revisions apported to the new version of the manuscript also meet the concerns that I have risen. In view of this, I can consider the manuscript "Programmable frequency-bin quantum states in a nano-engineered silicon device" by M. Clementi et al. eligible for publication in Nature Communications.

Reviewer #3 (Remarks to the Author):

The authors have clarified several aspects, making the paper suitable for Nature Communications.